# Fever in Children with Cancer: Pathophysiological Insights Using Blood Transcriptomics

**DOI:** 10.3390/ijms26157126

**Published:** 2025-07-24

**Authors:** Lotte Møller Smedegaard, Kia Hee Schultz Dungu, Yuliu Guo, Lisa Lyngsie Hjalgrim, Victoria Probst, Luca Mariani, Dorthe Grosen, Ines Kristensen, Ruta Tuckuviene, Kjeld Schmiegelow, Frederik Otzen Bagger, Nadja Hawwa Vissing, Ulrikka Nygaard

**Affiliations:** 1Department of Pediatrics & Adolescent Medicine, Copenhagen University Hospital, Rigshospitalet, 2100 Copenhagen, Denmark; lotte.moeller.smedegaard@regionh.dk (L.M.S.); kia.hee.schultz.dungu@regionh.dk (K.H.S.D.); lisa.lyngsie.hjalgrim@regionh.dk (L.L.H.); kjeld.schmiegelow@regionh.dk (K.S.); nadja.hawwa.vissing@regionh.dk (N.H.V.); 2Center for Genomic Medicine, Copenhagen University Hospital, Rigshospitalet, 2100 Copenhagen, Denmark; yuliu.guo@regionh.dk (Y.G.); victoriaprobst94@gmail.com (V.P.); luca.mariani@regionh.dk (L.M.); frederik.otzen.bagger@regionh.dk (F.O.B.); 3Department of Pediatrics & Adolescent Medicine, Hans Christian Andersen Children’s Hospital, 5000 Odense, Denmark; dorthe.grosen@rsyd.dk; 4Department of Pediatrics & Adolescent Medicine, Aarhus University Hospital, 8200 Aarhus, Denmark; inekri@rm.dk; 5Department of Pediatrics & Adolescent Medicine, Aalborg University Hospital, 9000 Aalborg, Denmark; ruta.tuckuviene@regionh.dk; 6Department of Clinical Medicine, Faculty of Health & Medical Sciences, Copenhagen University, 2200 Copenhagen, Denmark

**Keywords:** febrile neutropenia, pediatric oncology, transcriptomics, differential gene expression, bloodstream infection

## Abstract

Fever is a frequent complication in children receiving chemotherapy, primarily caused by bloodstream infections and non-infectious inflammation. Yet, the pathophysiological mechanisms remain unclear, and diagnostics are insufficient, which often results in continued antibiotic treatment despite negative blood cultures. In a nationwide study, we collected whole blood in PAXgene tubes from 168 febrile episodes in children with hematological malignancies, including 37 episodes with bacteremia, and performed single-cell RNA sequencing. We compared transcriptomic profiles between febrile children with and without bacteremia. In children with bacteremia, differentially expressed genes were related to immunoregulation and cardiac and vascular function. Children without bacteremia had distinct gene expression patterns, suggesting a viral or other inflammatory cause of fever. Several differentially expressed genes overlapped with previously published transcriptomics-based diagnostic signatures developed in immunocompetent children. In conclusion, blood transcriptomics provided novel insights into the pathophysiological mechanisms of febrile children with hematological malignancies. We found differentially expressed genes suggesting viral infections or non-bacterial inflammation as causes of fever in children with negative blood cultures, supporting early antibiotic discontinuation in children with cancer.

## 1. Introduction

Fever is a frequent and clinically important complication in children with cancer, most often observed during chemotherapy-induced neutropenia [1,2,3,4]. Bloodstream infections are the most common cause, accounting for 10–30% of fever episodes, and are associated with significant morbidity and mortality [5,6,7,8,9]. Other infections include viral infections, localized infections, such as skin infections related to central venous catheters, and, less commonly, fungal infections [10,11]. Non-infectious causes include cytokine release from chemotherapy-induced cell death, mucositis, transfusion reactions, and drug-induced fever, e.g., from cytarabine [12,13].

Distinguishing bloodstream infections from other causes of fever in children with cancer is essential to guiding targeted antibiotics and avoiding unnecessary treatment and hospitalization. Robust and generalizable tools for risk stratification are lacking, and current diagnostic methods remain slow or imprecise despite advances in pathogen-specific molecular techniques [14,15,16,17,18]. Early cessation of antibiotic therapy is now recommended in low-risk patients, but the duration in high-risk children without bacteremia but prolonged neutropenia remains a major research gap [19,20]. During the last decade, molecular technologies have enabled insights into the pathogenesis of the different causes of fever in immunocompetent children [21,22]. One method is transcriptomic profiling of whole blood, which detects activated genes in white blood cells and provides an unbiased analysis of biological processes, including immunological responses to infection [23,24]. Transcriptomics also holds promise for host-based diagnostics [22,25,26,27,28,29,30,31,32,33,34,35,36]. Although transcriptomics has posed challenges in children with cancer due to their low white blood cell counts [37,38], one study indicated that children with bacteremia have distinct immune responses from children with negative blood cultures [39].

Using transcriptomics, we aimed to investigate molecular changes in febrile children with hematological malignancies. We also aimed to assess the performance of a diagnostic two-gene signature, previously shown to yield promising results in immunocompetent children [26,28].

## 2. Results

The study included 168 episodes of fever from 93 children with hematological malignancies, comprising 56 boys (60%) and 37 girls (40%) with a median age of 5 years (IQR 3–10). Seventy-seven (83%) had acute lymphoblastic leukemia, eight (9%) had non-Hodgkin lymphoma, six had acute myeloid or bilineal leukemia (6%), and two (2%) had Hodgkin lymphoma (Figure 1).

Among the 168 fever episodes, 37 (22%) involved bacteremia, including 12 caused by *Enterobacterales* (*Escherichia coli*, *Klebsiella pneumoniae*, and *Enterobacter cloacae*), 8 by *Pseudomonas aeruginosa*, 7 by viridians group streptococci, 6 by *Staphylococcus aureus*, 2 by enterococci, 1 by *Stenotrophomonas maltophilia*, and 1 by *Acinetobacter pittii* (Appendix A). Of these, 12 had more than one pathogen identified. In six febrile episodes, coagulase-negative staphylococci were identified in one (n = 4) or two (n = 2) blood cultures. The levels of C-reactive protein (CRP), procalcitonin (PCT), white blood cell counts (WBC), and absolute neutrophil counts (ANC) are shown in Table 1.

### 2.1. Bacteremia Versus Unexplained Fever with Lov C-Reactive Protein

Transcriptomics revealed 266 differentially expressed genes in children with bacteremia compared to unexplained fever with low CRP levels (Figure 2A). Eleven of the top 20 differentially expressed genes were associated with immune response (Table 2).

Four genes were upregulated, including *C-X-C Motif Chemokine Ligand 1* (*CXCL1*), *Zinc and Ring Finger 1* (*ZNRF1*), *BTB Domain and CNC Homolog 2* (*BACH2*), and *Tumor Protein 53 Binding Protein 1* (*TP53BP1*). Several interferon-induced genes related to antiviral responses were downregulated, including *ISG15 Ubiquitin Like Modifier* (*ISG15*), *2*′-*5*′-*Oligoadenylate Synthetase 1* (*OAS1*), *2*′-*5*′-*Oligoadenylate Synthetase Like* (*OASL*), *Interferon Induced Protein With Tetratricopeptide Repeats 1, 2 and 3* (*IFIT1*, *IFIT2*, *IFIT3*)*,* and *Cytidine/Uridine Monophosphate Kinase 2* (*CMPK2*) (Table 2). In line, the gene set enrichment analysis identified two downregulated pathways related to antiviral responses, “*Antigen processing and presentation*” and “*Toll-like receptor signaling pathway*” (Figure 3A).

In addition, three of the top 20 differentially expressed genes were associated with cardiac and vascular function, including *Proprotein Convertase Subtilisin/Kexin Type 6* (*PCSK6*), *C-X-C Motif Chemokine Ligand 1* (*CXCL1*), and *Myozenin 3* (*MYOZ3*) (Table 2). In line, the gene set enrichment analysis identified upregulated pathways linked to cardiac and vascular function, such as “*Dilated cardiomyopathy*”, “*Hypertrophic cardiomyopathy*”, “*Vascular smooth muscle contraction*”, “*Regulation of actin cytoskeleton*”, and “*Calcium signaling pathway*” (Figure 3A). Several top differentially expressed genes were associated with cytoskeletal organization and cellular homeostasis (Table 2).

On day 2 of fever onset, 134 genes distinguished children with bacteremia from unexplained fever with low CRP levels. Only one gene, *SS18L1 Subunit of BAF Chromatin Remodeling Complex* (*SS18L1*), overlapped from day 1 to day 2. Among the top 20 differentially expressed genes, three related to immune regulation and neutrophil effector function, including *Cathepsin G* (*CTSG*), *myeloperoxidase* (*MPO*), and *Elastase, Neutrophil Expressed* (*ELANE*), were downregulated. The remaining were mainly involved in DNA and RNA processing and repair as well as protein modification and transport (Appendix A). Three distinct pathways were upregulated: “*Cytokine cytokine receptor interaction*”, “*Ribosome*”, and “*Taste transduction*” (Figure 3B). Three downregulated pathways overlapped with those from day 1, including “*Type I diabetes mellitus*”, “*Autoimmune thyroid disease*”, and “*Asthma*”.

### 2.2. Unexplained Fever with High C-Reactive Protein Versus Bacteremia

Comparing unexplained fever episodes with high CRP to those with bacteremia, 192 genes were differentially expressed (Appendix A). Four of the top 20 differentially expressed upregulated genes were immune-related, including *Interferon Alpha Inducible Protein 27* (*IFI27*)*, Fatty Acid Binding Protein 5* (*FABP5*), and *Chloride Channel 3* (*CLC*). *CXCL Motif Chemokine Ligand 1* (*CXCL1*) was downregulated together with differentially expressed genes linked to vascular dysfunction, including *Perlecan* (*HSPG2*) and *Myozenin 3* (*MYOZ3*). The remaining top differentially expressed genes were primarily related to mitochondrial energy production and function, RNA processing, and gene regulation (Appendix A). The gene set enrichment analysis identified downregulated pathways related to cardiac function, including “*Dilated cardiomyopathy*”, “*Hypertrophic cardiomyopathy*”, and “*Calcium signaling pathway*” (Figure 3C).

### 2.3. Coagulase-Negative Staphylococci vs. Bacteremia Caused by Gram-Negative and Other Gram-Positive Bacteria

No genes were differentially expressed in children with coagulase-negative staphylococci compared to bacteremia caused by Gram-negative and other Gram-positive bacteria. The gene set enrichment analysis revealed similar up- and downregulated pathways to those observed in unexplained fever with high CRP (Figure 3D).

### 2.4. Testing the Performance of a 2-Gene Signature

The *ADGRE1* and *IFI44L* 2-gene signature identified 14 (52%) of 27 episodes with bacteremia on day 1 as a bacterial infection (sensitivity 0.52, 95% CI 0.32–0.71) and 39 (60%) of 65 episodes without bacterial infection and low CRP as not having bacterial infections (specificity 0.60, 95% CI 0.47–0.72), using the proposed threshold of 0 (Figure 4). We could not identify an alternative risk score threshold as the groups had considerable overlaps in the disease risk score values (Figure 4).

## 3. Discussion

In this prospective cohort study, we demonstrated the diagnostic potential of transcriptomic profiling in children with hematological malignancies by identifying distinct gene expression patterns distinguishing children with bacteremia from those without. Children with bacteremia had increased expression of genes related to antibacterial immune responses, along with upregulated genes and pathways associated with cardiac and vascular function, potentially indicating sepsis. In contrast, children with unexplained fever had transcriptomic patterns dominated by interferon-related pathways, suggesting viral or other non-bacterial causes of fever. These results argue against false-negative blood cultures and support the potential for early discontinuation of antibiotics in these children despite persistent neutropenia.

This study is the largest to date investigating transcriptomic profiling in febrile children with cancer. Transcriptomic investigations of immunocompromised patients have posed challenges due to their low white blood cell counts, limiting the extraction of sufficient mRNA for gene expression analysis. Only two studies have investigated transcriptomic profiling in febrile children with cancer [37,39]. One study by Haeusler et al. found sufficient RNA for sequencing in 91% of peripheral blood mononuclear cells collected in EDTA tubes. However, this method requires immediate processing to preserve RNA, limiting its utility in clinical and research settings [39]. The other study by Wahlund et al. used RNA-stabilizing tubes, similar to our study, which preserve RNA at collection and allow storage at room temperature for up to 72 h before freezing. However, using a standard RNA-sequencing protocol, Wahlund et al. concluded that immunocompromised children had too few white blood cells for reliable gene expression analysis [37]. In our study, no samples were excluded due to low amounts of RNA, and 91% were successfully sequenced using a single-cell mRNA-sequencing protocol, which enhances mRNA capture and quantification [38].

In children with bacteremia, 266 genes were differentially expressed compared to children with unexplained fever and low CRP. This finding aligns with those of Haeusler et al., who found significant changes in gene expression were observed between children with bloodstream infection and unexplained fever [39]. In our study, several key immunoregulatory genes related to bacterial infection were upregulated, including *ZNFR1*, involved in *TLR4* signaling induced by Gram-negative bacteria; *BACH*, involved in the adaptive immune response; and *CXCL1*, a chemokine that promotes neutrophil recruitment and has recently been shown to increase earlier than CRP in children with cancer and bacteremia. Genes and pathways related to cardiac and vascular function were upregulated, such as *MYOZ3* and *HSPG2*, which may reflect hyperdynamic cardiac function and vascular dysregulation during sepsis. *CXCL1* may also contribute to vascular dysfunction during sepsis [23,40,41,42]. Pathways linked to calcium signaling were upregulated, as also found by Haeusler et al., consistent with its role in inflammatory activation, vascular permeability, cardiac muscle contraction, and coagulation [39]. Collectively, transcriptomic profiling at fever onset in children with bacteremia was characterized by innate and adaptive immune responses, cardiac and vascular function, and cytoskeletal organization.

A rapid change in gene expression was observed during the first 24 h of fever onset in children with bacteremia. Only one gene overlapped from day 1 to day 2, and no pathways were associated with cardiac or vascular function as seen on day 1. Several upregulated genes were related to cellular homeostasis, indicating a shift towards recovery [43,44]. Neutrophil-expressed genes were downregulated on day 2, reflecting a reduced innate immune response, which may suggest recovery or inappropriate immune suppression [43,45,46]. Thus, transcriptomic changes may vary during the infectious course. This dynamic RNA host response within 24 h may reflect physiological and immunological changes following treatment, such as antibiotics and fluid therapy [47]. Supporting this, Haeusler et al. [39] identified pathways involved in immune recovery and metabolic restoration on day 2, and Cazalis et al. [44] reported that over 70% of the leukocyte transcriptome was altered within 24 h in adults with septic shock, with substantial changes within the first hours. This highlights the importance of investigating transcriptomic changes when developing diagnostic signatures.

Bacteremia with coagulase-negative staphylococci is a common clinical dilemma, as it is often difficult to determine whether the organism is the cause of fever or merely a contaminant. We, therefore, analyzed these episodes separately. We found that coagulase-negative staphylococci had distinct pathways compared to Gram-negative and other Gram-positive infections, while most pathways overlapped with unexplained fever and high CRP. This suggests that coagulase-negative staphylococci are pathophysiologically different from other causes of bloodstream infection. Moreover, these findings may indicate that the coagulase-negative staphylococci were contaminants and that fever in these children was caused by a non-bacterial infection. However, the small sample size limits the ability to draw definitive conclusions.

Children with unexplained fever and low CRP had differentially expressed genes linked to interferon-induced immune responses, including *IFIT1*, *IFIT2*, and *IFIT3*, commonly upregulated during viral infections [48]. This suggests that children with unexplained fever and low CRP may have had fever due to (1) respiratory viral infections without focal signs, as respiratory viruses have been reported to be frequent in children with non-focal febrile neutropenic patients, (2) other non-focal infections, such as enterovirus, and/or (3) self-limiting reactivation of systemic viruses such as cytomegalovirus (CMV) and Epstein-Barr virus (EBV), although such reactivation has recently been reported to be rare in children with febrile neutropenia [49,50,51,52,53,54]. This remains speculative as we did not test for viral infections in children with non-focal febrile neutropenia. Alternatively, the observed interferon-response may reflect sterile inflammation due to non-infectious conditions, as described by Sweeney et al. [43], and thus potentially related to chemotherapy-induced tissue damage. If the observed downregulation of interferon genes in children with bacteremia is reflected at the protein level, the clinical implications could include a higher susceptibility to reactivation of latent viruses. This aligns with Cabler et al., who have reported that viremia, e.g., CMV and EBV, was frequent in pediatric sepsis and linked to a higher mortality [55].

Children with unexplained fever and high CRP had a gene expression profile distinct from those with bacteremia, including downregulated *CXCL1*, *MYOZ3*, and *HSPG2*. They had upregulated *IFI27*, a type-I interferon-induced gene commonly upregulated during viral infection and other inflammatory responses, and recently suggested as a target for immunoregulatory therapies [56,57,58,59,60]. The distinct profile argues against false negative blood cultures and supports the potential for shorter antibiotic courses despite persistent neutropenia. Shorter antibiotic courses in high-risk children with culture-negative febrile episodes remain a research gap and are currently being investigated in two randomized controlled trials [20,61,62].

Diagnostics based on transcriptomics have shown promise in immunocompetent children, whereas their diagnostic value in children with cancer remains uncertain. Neither the study by Haeusler et al. [39] nor ours included enough febrile episodes to develop and validate a diagnostic signature. The previously published 2-gene signature, based on *IFI44L* and *ADGRE1*, performed poorly in our cohort. Still, several differentially expressed genes identified in our study overlapped with those frequently appearing in other published signatures differentiating bacterial from non-bacterial infections developed in immunocompetent children, including *IFI27*, *OASL*, and *ISG15* [63,64]. This suggests that children with hematological malignancies may have select differentially expressed genes comparable to those of immunocompetent children, despite differences in immune cell composition. Thus, the finding supports the potential applicability of diagnostic signatures developed in immunocompetent populations to immunocompromised children. Perspectives involve investigations of other specific signatures developed in children [27], including innovative multi-class diagnostic signatures, designed to concurrently account for specific pathogens [32].

The main limitation of this study was the small sample size, which limited the investigation of the pathophysiology in specific bloodstream infections and hindered diagnostic signature development and validation. A second limitation was the lack of systematic testing for viral infections, potentially leading to undetected cases, as suggested by the gene expression analysis in children with unexplained fever. Third, as the day 1 and 2 samples were from different patients, sample variability limits direct comparison, unlike in Haeusler et al., where paired samples were used [39]. Still, both studies observed changes in gene expression over time. The main strengths included high RNA preservation, enabling successful sequencing with a single-cell mRNA-sequencing protocol, identification of immune genes overlapping with published signatures, and a study population encompassing the largest pediatric cancer cohort to date.

## 4. Materials and Methods

This nationwide observational study prospectively included children and adolescents aged 0–17 years with hematological malignancies, admitted to one of the four Pediatric Oncology Departments in Denmark from 1 June 2019 to 31 July 2021. Patients were eligible for inclusion if admitted with fever and re-inclusion if a new fever episode occurred. A new fever episode was defined as fever developing after termination of antibiotics or if the fever re-occurred during antibiotic therapy after at least 72 h of being afebrile (‘breakthrough fever’). Fever was defined as a temperature exceeding 38.5 °C or between 38 °C and 38.5 °C measured twice within one hour. Febrile episodes excluded localized infections, such as skin infections, possible blood culture contaminants, e.g., *Micrococcus* and *Corynebacterium* species, and confirmed viral infections. Patients without focal signs were not tested for respiratory viruses.

The patients were managed according to the Danish guidelines for fever in children with cancer, which included the administration of antipseudomonal antibiotics after blood cultures were obtained. They received anti-leukemic treatment according to the Nordic Society of Paediatric Hematology and Oncology (NOPHO) protocols, which did not include antibacterial prophylactic treatment except for weekly co-trimoxazole.

Whole blood (2.5 mL) for RNA analysis was collected at admission for fever, or within 24 h, in PAX gene^®^ Blood RNA Tubes (cat. no. 762165, Qiagen, Hilden, Germany) together with blood cultures and routine blood biochemistry. The samples were incubated at room temperature for at least two hours and frozen to −20 °C within 72 h. Demographic and clinical data were collected in real time from the electronic medical records into a REDCap database.

### 4.1. Definitions

RNA samples were defined as "day 1" if collected at admission due to fever, before initiating antibiotics, or at the time of fever onset in a patient receiving antibiotics. RNA samples collected 8–24 h after fever onset (on the following day) were defined as “day 2” samples.

We grouped febrile episodes by blood culture status and CRP levels, as those with high CRP and negative blood cultures remain clinically ambiguous, potentially reflecting culture-negative bacteremia, viral infection, or hyperinflammation. Similarly, febrile episodes with coagulase-negative staphylococci bacteremia were grouped separately. Thus, the cohort was divided into four groups:Bacteremia: Positive blood cultures, excluding coagulase-negative staphylococciUnexplained fever with low CRP: Negative blood culture and a maximum CRP below 35 mg/LUnexplained fever with high CRP: Negative blood culture and a maximum CRP of 35 mg/L or aboveCoagulase-negative staphylococci

### 4.2. RNA Sequencing and Quality Control

A total of 185 samples were analyzed, with 19 samples processed according to a previously published protocol [38], and 166 using the same protocol with the following minor modifications: Total RNA was isolated from stabilized blood using the QIAsymphony PAXgene Blood RNA Kit (Qiagen, Cat. Nr. 762635), according to the manufacturer’s instructions, and eluted in 100 µL. RNA concentration was quantified by NanoDrop. As RNA yields were within the recommended working range of the SMART-Seq HT Kit (Takara Bio, Shiga, Japan, Cat. Nr. 634436), the Eppendorf Concentrator Plus step used in the previously published protocol was omitted [38]. Samples with an RNA concentration above 12 ng/µL (group 1) and 5–12 ng/µL (group 2) were diluted to 1 ng/µL and 1:2, respectively, and samples with an RNA concentration below 5 ng/µL (group 3) were not further diluted. High-quality, full-length cDNA was generated by priming mRNA with oligo(dT) using the SMART-Seq HT Kit, as per the manufacturer’s instructions. The amount of mRNA used as input was standardized as follows: 1 µL of diluted mRNA for samples of group 1; 5 µL of diluted mRNA for samples of group 2; 10.5 µL of undiluted mRNA for samples of group 3. cDNA was amplified for 13 total PCR cycles and purified with AMPure XP beads (Beckman Coulter Life Sciences, Brea, CA, USA, A63881). cDNA concentration was measured with a Qubit Fluorometer (Thermo Fisher Scientific, Waltham, MA, USA, Q32854), and cDNA was diluted to 0.2 ng/µL, tagmented and barcoded using the Nextera XT DNA Library Preparation Kit (Illumina, San Diego, CA, USA, Cat. Nr. FC-131-1096), as per manufacturer’s instructions. Final libraries were purified with AMPure XP beads. DNA concentration was measured with a Qubit Fluorometer, and fragment size was determined with a Tape Station using High Sensitivity D5000 reagents (Agilent, Santa Clara, CA, USA, Cat. Nr. 5067-5593). Samples were paired-end sequenced on a NextSeq 500 Sequencer (Illumina) to reach a sequencing depth of 1 million raw reads per sample.

The Illumina raw reads were exported as fastq files and trimmed using TrimGalore! v.0.4.0 [65]. The trimmed sequences were aligned to the Genome Reference Consortium Human Build 38 (GRCh38) by STAR (Spliced Transcripts Alignment to a Reference) [66]. For quality control, principal component analysis was performed to check for batch effect, mitochondrial content, age, and gender. Low-quality samples were defined as those with expressed genes less than or equal to 5000, a number of unmapped reads greater than 2^18^ (262,144), and mitochondrial genes exceeding 30%. Seventeen (9%) of 185 samples were excluded, including 6 lost during the processing steps and eleven excluded after quality control, resulting in a final study population of 168 febrile episodes.

### 4.3. Differential Expression and Pathway Analysis

The primary analysis compared differentially expressed genes between febrile episodes with bacteremia and unexplained fever with low CRP on day 1, followed by a comparison of the same groups on day 2. Additionally, separate analyses compared unexplained fever with high CRP to febrile episodes with bacteremia and episodes with coagulase-negative staphylococci to those with bacteremia. Raw RNA-seq count data were imported into RStudio (v4.3.3). Differential gene expression analysis was performed using DESeq2 (v1.37.0), with neutropenia status (yes/no), febrile infection episodes, and RNA sample collection time considered in the design formula. Genes with fewer than 10 total counts across all samples were filtered out before performing differential expression analysis. Differentially expressed genes (DEGs) were identified with a threshold of absolute value of log_2_FoldChange(log_2_FC) more than 1.5, and adjusted *p*-value (padj) less than 0.05. Volcano plots were used to visualize differentially expressed genes. The function of differentially expressed genes was obtained from GeneCards [67].

Gene set enrichment analysis (GSEA) was performed using fgsea (v1.30.0) package, with log_2_FC as the ranking metric. Significantly enriched pathways were determined based on padj through Kyoto Encyclopedia of Genes and Genomes (KEGG) pathway analysis, uncovering biological processes associated with differentially expressed genes.

To investigate the performance of a previously published 2-gene signature [26,28], a disease risk score was calculated as the log2-transformed expression difference between ADGRE1 and IFI44L, using a threshold for bacterial assignment of above 0, as proposed by Pennisi et al. [26]. Expression values were extracted from DESeq2-normalized counts, and log2-transformed values were computed with a small offset (+0.1) to prevent log transformation errors. The scores were visualized using violin plots and sina plots [68,69].

### 4.4. Study Approvals

Patients were recruited under the approval of the research ethics committees of the Ethics Committee of the Capital Region of Denmark (H-2-2010-002) and the Danish Data Protection Agency (P-2019-29). Informed oral and written parental consent was provided before participation.

## 5. Conclusions

In conclusion, this study provided novel insights into the pathophysiological mechanisms in febrile children with hematological malignancies, revealing immunoregulatory genes associated with bacterial infection and genes and pathways related to cardiac and vascular function, aligning with sepsis. Children with unexplained fever had gene expression profiles different from those with bacteremia, including several interferon-induced genes, suggesting a viral or non-bacterial cause of inflammation, and challenging the assumption of false-negative blood cultures. This finding supports early discontinuation of antibiotics in children with unexplained fever. Furthermore, the study identified several genes overlapping with previously published signatures, supporting the potential for developing transcriptomics-based diagnostics in children with cancer.

## Figures and Tables

**Figure 1 ijms-26-07126-f001:**
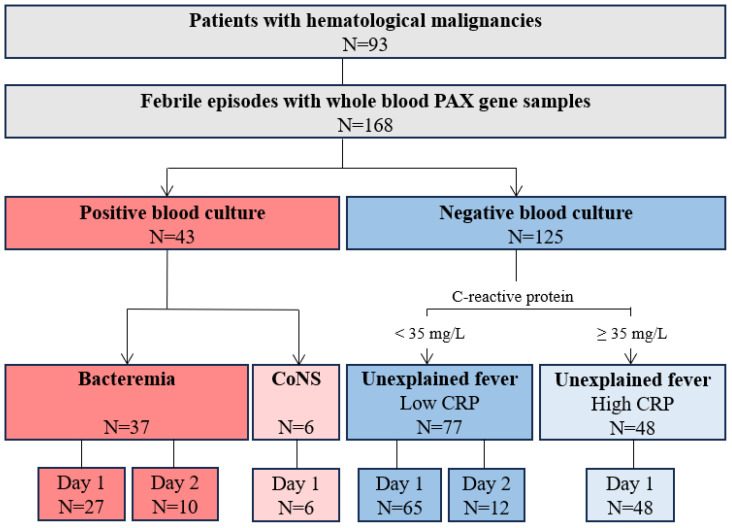
Classification of febrile episodes. Febrile children were classified according to blood cultures and CRP. The primary comparison was between febrile episodes with bacteremia and unexplained fever with low CRP. Seventeen (9%) samples from 185 febrile episodes were excluded (6 during processing steps and 11 after quality control), resulting in a final study population of 168 episodes. CoNS = Coagulase-negative staphylococci; CRP = C-reactive protein.

**Figure 2 ijms-26-07126-f002:**
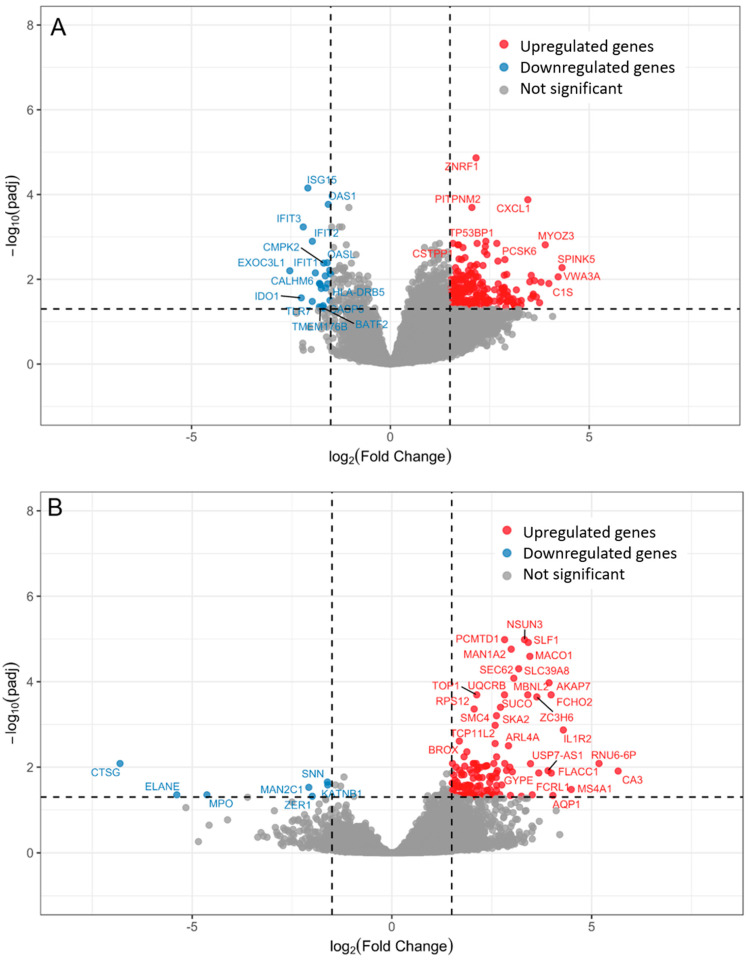
Volcano plot illustrating differentially expressed genes between febrile episodes with bacteremia and unexplained fever with low CRP on day 1 (**A**) and day 2 (**B**). Each point represents a gene. The x-axis represents the log_2_ fold change, and the y-axis represents the log_10_ (*p*-value). The dashed lines represent the statistical significance thresholds after adjusting for multiple testing. Upregulated genes in children with bacteremia compared to unexplained fever with low CRP-reactive protein are shown in red, such as *C-X-C Motif Chemokine Ligand 1* (*CXCL1*), and downregulated in blue, such as *ISG15 Ubiquitin Like Modifier* (*ISG15*). Genes that did not differ significantly between groups are gray. One gene overlapped between the day 1 and day 2 samples.

**Figure 3 ijms-26-07126-f003:**
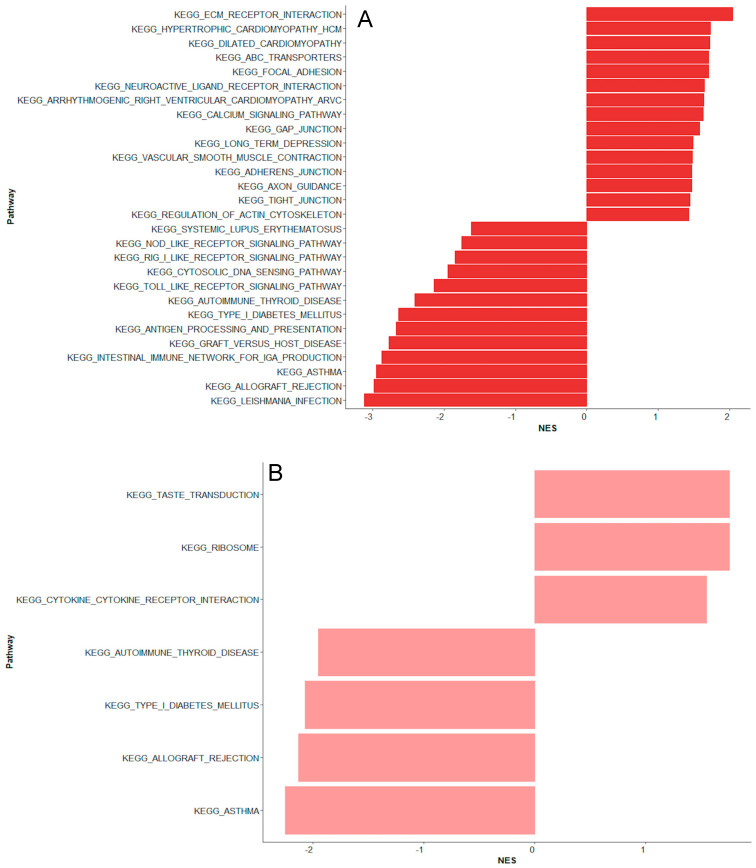
Gene enrichment pathways. Biological process enrichment analysis of differentially expressed genes across four comparisons: (**A**) Bacteremia day 1 versus unexplained fever with low CRP, (**B**) bacteremia day 2 versus unexplained fever with low CRP day 2, (**C**) unexplained fever with high CRP day 1 versus bacteremia day 1, and (**D**) coagulase-negative staphylococci day 1 versus bacteremia day 1. Pathways were identified through Kyoto Encyclopedia of Genes and Genomes (KEGG) analysis and displayed by Normalized Enrichment Score (NES), which reflects the direction and strength of enrichment. NES > 0 indicates upregulated pathways and NES < 0 indicates downregulated pathways in the first group of each comparison. NES are shown on the x-axis, and gene set names are on the y-axis. NES = Normalized Enrichment Score.

**Figure 4 ijms-26-07126-f004:**
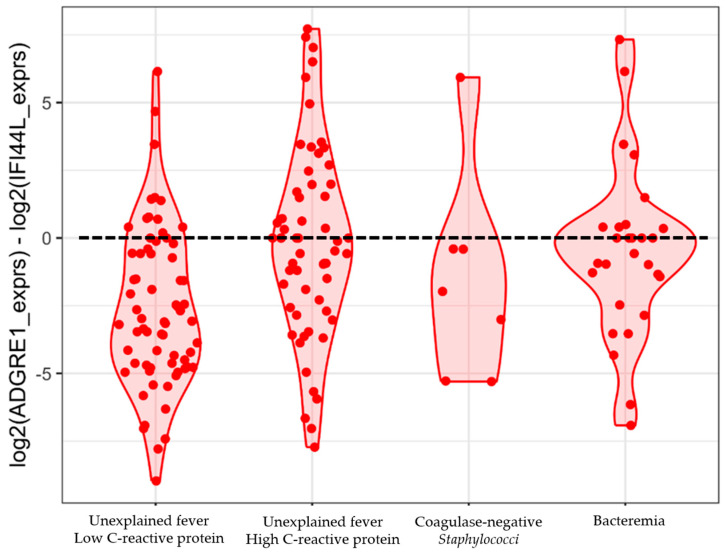
The performance of the previously published 2-gene signature, including *ADGRE1* and *IFI44L*, for unexplained fever with low and high CRP protein, coagulase-negative staphylococci, and bacteremia. To investigate the performance of the 2-gene signature, a disease risk score for each febrile episode was calculated by subtracting log2 *IFI44L* expression from log2 *ADGRE1* expression. The dotted line represents the proposed threshold of 0 or above for assignment to bacterial infections [25,26]. All calculations used febrile episodes on day 1.

**Table 1 ijms-26-07126-t001:** Hematological and inflammatory parameters in patients with febrile episodes.

Febrile Episodes	WBC ×10^9^ cells/L	ANC ×10^9^ cells/L	CRP mg/L	PCT ng/mL
**Positive blood culture**				
Bacteremia				
- Day 1	0.5 (0.1–2.4)	0.0 (0.0–1.4)	84 (40–130)	5.9 (0.7–15.9)
- Day 2	0.3 (0.1–0.5)	0.0 (0.0–0.0)	200 (150–287)	4.3 (0.4–95.0)
Coagulase-negative staphylococci	1.5 (0.5–2.3)	0.8 (0.0–1.6)	58 (17–115)	0.5 (0.1–1.3)
**Negative blood culture**				
C-reactive protein < 35 mg/L				
- Day 1	3.6 (1.6–5.9)	2.5 (0.8–4.6)	7 (4–20)	0.2 (0.1–0.3)
- Day 2	1.6 (1.1–3.2)	0.3 (0.1–1.1)	19 (8–25)	0.2 (0.1–0.3)
C-reactive protein ≥ 35 mg/L	0.7 (0.3–0.6)	0.0 (0.0–0.3)	88 (54–137)	0.7 (0.3–1.5)

ANC = Absolute Neutrophil Count; CRP = Absolute Neutrophil Count; PCT = Procalcitonin; WBC = White Blood Cell Count.

**Table 2 ijms-26-07126-t002:** Differentially expressed genes in febrile episodes with bacteremia vs. unexplained fever with low C-reactive protein on day 1.

Gene Symbol	Gene Name	log_2_ FC	*p* _adj_	Gene Function
**Immune response and regulation**				
*CXCL1*	*C-X-C Motif Chemokine Ligand 1*	3.46	0.0001	Chemotactic activity for neutrophils
*ZNRF1*	*Zinc and Ring Finger 1*	2.16	<0.0001	Regulation of inflammation, TLR4-activated immune response, controlling TLR3 trafficking
*BACH2*	*BTB Domain and CNC Homolog 2*	1.72	0.0015	Regulates T-cell function, B-cell maturation, and inflammatory response
*TP53BP1*	*Tumor Protein 53 Binding Protein 1*	2.40	0.0013	DNA repair in lymphocytes; class switch recombination in B cells
*ISG15*	*ISG15 Ubiquitin-Like Modifier*	−2.07	<0.0001	Interferon-induced; essential in innate antiviral immune response
*OAS1*	*2′-5′-Oligoadenylate Synthetase 1*	−1.56	0.0002	Interferon-induced; essential in innate antiviral immune response
*IFIT3*	*Interferon-Induced Protein with Tetratricopeptide Repeats 3*	−2.19	0.0006	Interferon-induced antiviral protein; inhibitor of cellular and viral processes, cell migration, proliferation, signaling, and viral replication
*IFIT2*	*Interferon-Induced Protein with Tetratricopeptide Repeats 2*	−1.96	0.0013	Interferon-induced antiviral protein; inhibits expression of viral messenger RNAs
*OASL*	*2′-5′-Oligoadenylate Synthetase Like*	−1.59	0.0041	Negative regulation of viral genome replication, IL-27-mediated signaling pathway
*CMPK2*	*Cytidine/Uridine Monophosphate Kinase 2*	−1.68	0.0042	Immunomodulatory and antiviral activities
*IFIT1*	*Interferon-Induced Protein with Tetratricopeptide Repeats 1*	−1.89	0.0071	Interferon-induced antiviral protein; inhibits viral replication
**Other**				
*SS18L1*	*SS18L1 Subunit of BAF Chromatin Remodeling Complex*	1.70	0.0015	Calcium-responsive transactivator; an essential subunit of a neuron-specific chromatin-remodeling complex
*CSTPP1*	*Centriolar Satellite-Associated Tubulin Polyglutamylase Complex Regulator 1*	1.58	0.0014	Cytoskeletal organization, nuclear shape, and cilium disassembly
*PITPNM2*	*Phosphatidylinositol Transfer Protein Membrane Associated 2*	2.05	0.0002	Calcium ion binding and lipid binding
*MYOZ3*	*Myozenin 3*	3.90	0.0015	Skeletal and cardiac muscle; calcineurin-interacting, helps tether calcineurin to the sarcomere
*ARHGEF9*	*Cdc42 Guanine Nucleotide* *Exchange Factor 9*	2.18	0.0014	Brain-specific; receptor recruitment in GABAergic and glycinergic synapses
*DUSP6*	*Dual Specificity Phosphatase 6*	−1.51	0.0075	Inactivates MAP kinases associated with cellular proliferation and differentiation
*PCSK6*	*Proprotein Convertase* *Subtilisin/Kexin Type 6*	2.68	0.0014	Vascular remodeling; processes protein and peptide precursors in the liver, gut, and brain
*ODF3B*	*Outer Dense Fiber of Sperm Tails 3B*	−1.52	0.0061	Functional element
*EXOC3L1*	*Exocyst Complex Component 3 Like 1*	−2.53	0.0063	Part of the exocyst; possible role in regulating exocytosis of insulin granules

Gene functions were retrieved from https://www.genecards.org/.

## Data Availability

The datasets presented in this article are not readily available in adherence to guidelines established by the Danish Data Protection Agency, which classifies RNA sequences as personally identifiable information. Requests to access the datasets should be directed to the corresponding author.

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
