# Peer review of "Fever in Children with Cancer: Pathophysiological Insights Using Blood Transcriptomics"

_ijms, 2025, doi:10.3390/ijms26157126_

Round 1
Reviewer 1 Report
Comments and Suggestions for Authors
The study used a transcriptomics analytic approach in children with malignancies (mainly ALL) and febrile neutropenia, with different subsets of bacteriological and inflammatory situations.
RNA was successfully extracted from blood samples despite a low count of neutrophils.
Expressed genes are different for patients with bacteremia (expressed genes related to immunoregulation and cardiac/vasculare function) and other situations such as viral infections and inflammation with distinct gene expression patterns.
This study appears as the most large study of blood transcriptomics and brings new insights on the physiopathological mechanisms of the fever in children with neutropenia.
The size of samples (collected from 2019 to 2021) did not allow to perform a diagnostic RNA signature.
Very interesting paper, with datas supporting the early discontinuation of antibiotics in the situation of febrile neutropenia
I wonder why the Material and Methods section appears after the Discussion section. No other concerns.
References are adequate and complete.
Reviewer 2 Report
Comments and Suggestions for Authors
Here, the authors use blood transcriptomics to reveal the underlying pathophysiological mechanisms of fever in pediatric cancer patients, particularly during episodes of chemotherapy-induced neutropenia. By analyzing 168 fever episodes in children with hematological malignancies, the study distinguishes between cases of bacteremia and those with unexplained fever accompanied by low C-reactive protein levels. In episodes of bacteremia, transcriptomic profiling revealed an increase in genes associated with antibacterial immune responses as well as pathways linked to cardiac and vascular functions, which may indicate the early stages of sepsis. In contrast, fever episodes without a positive blood culture were characterized by the upregulation of interferon-related genes, suggesting a viral etiology or a sterile inflammatory process rather than a bacterial one. Additionally, the article assessed the performance of a previously validated two-gene diagnostic signature based on ADGRE1 and IFI44L; however, this signature did not reliably classify infections in the present cohort of immunocompromised children. The article is well written, with figures of adequate quality and robust data analysis.
Reviewer 3 Report
Comments and Suggestions for Authors
The present study is methodologically strong, clinically relevant, and biologically coherent, and it lays the groundwork for future development of transcriptomics-based diagnostics in immunocompromised children. The study presents valuable pathophysiological insights into febrile episodes in immunocompromised pediatric patients, leveraging single-cell RNA sequencing. However, there are some minor weaknesses that need to be addressed before this research article becomes acceptable for publication.
Major Comments
- The interpretation of immune-related pathways could benefit from additional biological context. Specifically, a more detailed discussion of the clinical implications of downregulated interferon-stimulated genes (e.g., IFIT1-3, ISG15) would enhance clarity.
- The manuscript shows interesting temporal dynamics between days 1 and 2. Nonetheless, only one gene overlapped between both timepoints. The authors should comment on the potential causes of this low overlap (e.g., treatment effect, sample variability).
- The sensitivity (52%) and specificity (60%) of the 2-gene signature are modest. Consider recommending a comparison with other published diagnostic signatures.
- The absence of differentially expressed genes between CoNS and other bacteremias is noteworthy. Given the clinical controversy surrounding CoNS as true pathogens versus contaminants, the authors should discuss whether transcriptomic silence supports the latter interpretation.
Minor Comments
- The figures, particularly the volcano plots and enrichment analyses (Figures 2 and 3), are difficult to read due to small font sizes and low contrast.
- Several figure legends (e.g., Figures 2 and 4) could be more informative.
- In some places, “unexplained fever with low CRP” is used, and elsewhere “controls with low CRP” or just “low CRP group” is written.
- There are several minor grammatical and typographical errors throughout the text. Careful proofreading is recommended.
- Table 1 would benefit from clearer separation of groups and perhaps color-coding or shading to distinguish positive vs. negative cultures more effectively.
